# Parkinson’s Disease Dementia Patients: Expression of Glia Maturation Factor in the Brain

**DOI:** 10.3390/ijms25021182

**Published:** 2024-01-18

**Authors:** Ramasamy Thangavel, Harleen Kaur, Iuliia Dubova, Govindhasamy Pushphavathi Selvakumar, Mohammad Ejaz Ahmed, Sudhanshu P. Raikwar, Raghav Govindarajan, Duraisamy Kempuraj

**Affiliations:** Department of Neurology, Center for Translational Neuroscience, School of Medicine, University of Missouri, Columbia, MO 65212, USA; ramasamy-thangavel@uiowa.edu (R.T.); hkaur44@northwell.edu (H.K.); idubova@mgh.harvard.edu (I.D.); gp-selvakumar@uiowa.edu (G.P.S.); mahmed14@hfhs.org (M.E.A.); sudhanshu.raikwar@barrowneuro.org (S.P.R.); raghav.govindarajan@hshs.org (R.G.)

**Keywords:** Alzheimer’s disease, amyloid plaques, dopaminergic neurons, glial fibrillary acidic protein, glia maturation factor, ionized calcium-binding adaptor molecule 1, neurofibrillary tangles, Parkinson’s disease dementia, substantia nigra, tyrosine hydroxylase

## Abstract

Parkinson’s disease (PD) is the second most common progressive neurodegenerative disease characterized by the presence of dopaminergic neuronal loss and motor disorders. PD dementia (PDD) is a cognitive disorder that affects many PD patients. We have previously demonstrated the proinflammatory role of the glia maturation factor (GMF) in neuroinflammation and neurodegeneration in AD, PD, traumatic brain injury (TBI), and experimental autoimmune encephalomyelitis (EAE) in human brains and animal models. The purpose of this study was to investigate the expression of the GMF in the human PDD brain. We analyzed the expression pattern of the GMF protein in conjunction with amyloid plaques (APs) and neurofibrillary tangles (NFTs) in the substantia nigra (SN) and striatum of PDD brains using immunostaining. We detected a large number of GMF-positive glial fibrillary acidic protein (GFAP) reactive astrocytes, especially abundant in areas with degenerating dopaminergic neurons within the SN and striatum in PDD. Additionally, we observed excess levels of GMF in glial cells in the vicinity of APs, and NFTs in the SN and striatum of PDD and non-PDD patients. We found that the majority of GMF-positive immunoreactive glial cells were co-localized with GFAP-reactive astrocytes. Our findings suggest that the GMF may be involved in the pathogenesis of PDD.

## 1. Introduction

Parkinson’s disease (PD) is a progressive, multi-symptom, neurodegenerative movement disorder in older adults [1,2]. PD is the second most common neurodegenerative disease after Alzheimer’s disease (AD) [3]. The clinical motor features in PD patients are tremors, rigidity, and bradykinesia. Neurochemically, the disappearance of melanin-containing dopaminergic (DA) neurons and the presence of neuroinflammation in the substantia nigra (SN) are major hallmarks of PD. PD affects nearly one million people in the United States (U.S) and ten million people worldwide (The Parkinson’s Foundation, Miami, FL, USA) [4]. By 2030, the number of PD patients is expected to increase to 1.2 million in the U.S. [4]. About 90,000 cases are diagnosed with PD each year in the U.S [4]. Men are about 1.5 time more prone to develop PD than women [4]. PD patients show cognitive impairment [2,5,6,7]. About 60 to 80% of PD patients may eventually show Parkinson’s disease dementia (PDD) [8]. Age is the primary risk factor for PD pathogenesis. PD pathogenesis may be due to certain factors such as environment, genetics, and exposure to toxins [3,9]. Alpha-synuclein aggregation, oxidative stress, mitochondrial abnormalities, immune dysfunction/neuroinflammatory response, and gut dysbiosis are implicated in PD pathogenesis [1,3,10,11]. Memory disorders are associated with hippocampal cholinergic system disorder [6]. A recent report proposed the involvement of hippocampus, striatum, and amygdala in PD [6]. Another recent study reported synaptic degeneration in temporal, cingulate, and insular cortices in PD [12], as well as in parahippocampus in PDD. Dementia is a brain disorder with loss of clear thinking, loss of memory, difficulty in making decisions, and difficulty [6] in controlling emotions that interfere with a person’s daily essential activities. AD is the most common cause of dementia in the aged population [13]. However, dementia may also be due to other diseases such as PD and Huntington’s disease. Dopamine is the neurotransmitter in dopaminergic neurons, and disturbances in dopamine lead to dopaminergic neurodegeneration in PD [14,15]. In addition, the loss of DA neurons in PD may be due to necrotic processes, though many other previous studies have shown that the loss of DA cells in PD patients is a result of apoptotic mechanisms [16,17,18]. Dementia in PD patients may significantly impact those with AD since it affects both the motor and cognitive functions in PD patients. AD affects language and memory, and PD involves problem solving, speed of thinking, memory, and other cognitive functions, including mood changes. A recent article links neuroinflammation with neurovascular changes in synucleinopathies [19].

Genetic factors like A beta, tau, APOE4, Presenilin, and TREM2 gene induce alterations in AD and α-Synuclein, Parkin, and LRRK2 gene associations in PD. A growing number of findings imply that defects in gene expression play a crucial role in neurodegeneration in the brain, specifically in AD and PD progression. Those include amyloid beta precursor protein (APP) on chromosome 21 [20]. And Presenilin (PSEN) 1 and 2 on chromosome 14 and 1 [21,22,23]. However, PSEN1 is the most mutated gene that is associated with the majority of cases of the onset of AD. Although those gene defects are suspicions, more genetic findings suggest the role of both familial and sporadic forms in PD as well as in AD. These five mutated genes have been shown to trigger familial forms of early-onset PD; these include α-synuclein, Park1 or SNCA, [24] Park2, [24] Park7, PINK1, [25] and LRRK2 [26,27]. In addition to that, six additional gene mutations at PD loci have been reported recently; those include UCHL1 and Nurr [28,29] at chromosome 4p14 and 2q22.

Previous studies have shown that the glia maturation factor (GMF) plays an important role in the pathogenesis of AD [30,31,32]. Numerous studies have demonstrated that the inflammatory processes in PD brains are directly correlated with the activation of both microglial cells and astrocytes [33,34,35,36,37]. Microglial activation leads to the release of proinflammatory cytokines and chemokines, including the GMF in the brain. The GMF is a brain-specific protein that was discovered, isolated, sequenced, and cloned previously in our laboratory [38,39,40]. We have previously demonstrated that the GMF is an important mediator of inflammation in the central nervous system (CNS), which can activate glial cells to release inflammatory mediators and induce neurodegeneration in neurodegenerative diseases as well as stroke and neurotrauma conditions [41,42,43]. PD patients show cognitive impairment from early to more advanced stages [44]. The GMF could be involved in the pathogenesis of PDD. If the degeneration of DA neurons in SN was associated with GMF up-regulation in PDD, it would suggest that the inhibition of GMF expression through anti-inflammatory therapy such as GMF-specific monoclonal antibodies or GMF gene silencing approaches including small interfering RNA (siRNA), short hairpin RNA (shRNA), and CRISPR/Cas9 based GMF gene editing for PD patients might have beneficial effects by halting the neurodegenerative process or inhibiting neuroinflammatory mechanisms. AD is characterized by the accumulation of amyloid plaques (Aps) and neurofibrillary tangles (NFTs) in the brain. NFT pathology extends anatomically throughout various interconnected regions of the brain [45,46]. The deposition of Aps in the brain occurs through several phases. APs and NFTs are formed in the later stages of the disease, and they are located in closely interconnected areas of the SN and striatum and can precipitate extrapyramidal motor symptoms in patients with AD [47]. The pathophysiological changes seen in advanced stages of neurodegenerative disorders, like AD and PD, are a result of neuroinflammatory responses by activated astroglial cells. We hypothesize that neurodegeneration due to the increased secretion of proinflammatory cytokines from activated astroglial cells by the up-regulation of the GMF in the most affected regions of the SN could contribute to the development of dementia in PD.

Activated microglia and astrocytes are increased and prominent in the damaged regions of the SN and play important roles in PD pathogenesis [48,49,50]. However, studies have also shown reduced reactive astrocytosis in the SN of PD [33,34]. In the present study, we report an increased expression of the GMF in the brains of PDD patients. In addition, the GMF was found to be co-localized with glial fibrillary acidic protein (GFAP) reactive astrocytes, Aps, and NFTs in the SN of PDD patients.

## 2. Results

### 2.1. Decreased TH and Increased α-Synuclein and Ubiquitin Expression in the SN of PDD Brains

To assess Lewy body (LB) pathology and dopaminergic neurodegeneration, IHC for TH was performed in the SN of PDD brains (Figure 1). Results show that the expression of TH (immunoreactivity) was reduced in PDD compared with non-PDD brains (Figure 1). Immunostaining with anti-ubiquitin and anti-α-synuclein antibodies showed more LBs containing α-synuclein and ubiquitin protein in PDD brains (Figure 1, arrowheads), demonstrating PD characteristics. The average number of TH, α-synuclein, and ubiquitin-positive cells is shown in the bar graphs (* *p* < 0.05). Next, we performed thioflavin-S fluorescence staining to detect the presence and extent of APs deposition and NFTs in the brains of patients with PDD and the non-PDD controls (Figure 2). We found an increased visualization of APs (Figure 2A; arrows) and NFTs (Figure 2A, arrowhead) in the SN containing dopaminergic neurons in PDD brains, indicating the severity of disease characteristics in the affected brains. However, non-PDD brains showed neither APs nor NFTs (Figure 2B). Graphs show the average numbers of NFTs (Figure 2C) and Aps (Figure 2D) in the SN of PDD and non-PDD brains (* *p* < 0.05).

### 2.2. GMF Is Associated with Glial Activation in the SN of PDD Brains

Next, we examined if GMF expression is associated with activated glial cells in the SN of PDD patients. Immunostaining results show melanin-containing dopaminergic cells in the SN of PDD and non-PDD brains (Figure 3A). We demonstrate the presence of GMF (immunoreactive astrocytes) in the vicinity of the dopaminergic neurons (Figure 3A, blue–gray color). Higher densities of GMF immunoreactive cells were seen in PDD brains, as shown in the graph (Figure 3B, * *p* < 0.05). The expression of immuno-positive astrocytes was detected near dopaminergic neurons in PDD brains (Figure 3A). In addition, we show an increased expression of Iba-1-positive microglia near the neuromelanin-containing cells in the SN of PDD brains. Our results indicate that the GMF induces glial activation-associated dopaminergic neuronal death in the SN of PDD patients.

### 2.3. Tau and Amyloid Levels in the SN, Striatum, HC, EC, and TC of PDD Brains

We performed immunostaining for Tau and Aβ for NFTs, plaques, and thioflavin-S fluorescence staining in the midbrain, striatum, and TC of PDD brains. PDD had tau NFTs and APs pathology in the striatum, SN, HC, EC, and TC of the brain (Figure 4). Tau immunostained NFT pathology and neuropil threads were seen in the SN, striatum, HC, EC, and TC of PDD brains. The marked expression of Tau-stained NFTs in the non-dopaminergic neurons in the SN of PDD was noted. 6E10 immunostained sections showed diffuse amyloid pathology in the striatum and SN. We observed a variable number of large, round, and diffuse APs in both the striatum and SN of PPD. Numerous small dense and diffuse APs are strongly stained aggregates of Aβ. We performed Tau immunostaining in the SN and striatum and used SG-vector substrate (blue–gray color) to detect the expression of NFTs and APs (Figure 4). We observed the marked expression of NFTs localized near neuromelanin-containing cells (more susceptible to death) stained in a brown color (Figure 4 arrowheads, arrows). Thioflavin-S (Thio-S) staining of the SN displayed the increased expression of APs (Figure 4). Tau immunostaining localized numerous NFTs in the striatum, SN, HC, EC, and TC of PDD brains. With 6E10 antibody immunolabeling, we show an increased expression of APs in the striatum, EC, HC, and TC of PDD brains. Overall, our results indicate that several NFTs and APs are localized in the SN and striatum of PDD brains.

## 3. Discussion

Dementia is a major health problem, and the cognitive disorder is an important non-motor dysfunction in PD patients [8,44,51]. PDD and dementia with LBs (DLB) together represent the second most common dementia across the world [52]. The cognitive disorders in PDD and DLB are different from Alzheimer’s dementia, with significantly impaired executive function, attention, and visuospatial skills [53]. AD co-pathology occurs with PD, especially in PDD and dementia with DLB [54]. Both PDD and DLB show cognitive dysfunction and α-synuclein deposition. In PDD, cognitive symptoms develop more than a year after the onset of movement symptoms. Vascular disorders affect cognition, which leads to dementia and cognitive decline [55]. The presence of excessive Aβ, which shows great sensitivity to differentiate between DLB and PDD, has been extensively investigated [56,57]. The burden of Aβ and tau proteins follow PD < PDD < DLB [58]. Immune and mitochondrial dysfunction, increased BBB permeability, and increased inflammatory mediators play an important role in PD pathogenesis [1]. Immune dysfunction and neuroinflammation are implicated in PD pathogenesis [10,11,59,60,61,62,63]. The GMF is a proinflammatory mediator that activates astrocytes, microglia, and neurons and induces neuroinflammation and neurodegeneration in many neurodegenerative conditions and neurotrauma through releasing several neuroinflammatory and neurotoxic mediators and NF-kB pathway activation [30,64,65,66,67]. In the present study, we demonstrate an increased expression of GMF in the SN of PDD brains. The GMF antibody used in our study detects a beta form of protein, mainly astrocytes and neurons. Increased levels of GMF in PDD indicate an inflammatory response, and further in-depth studies are needed to discover the specificity of the GMF in PDD brains. In the areas of increased GMF immunoreactivity, reactive astrogliosis was also observed along with increased microglial activation near the dopaminergic neurons in the SN region. Microglial activation is considered to be a marker of the neuroinflammatory process in PD and PDD [68,69,70], as well as other α-synucleinopathies and multiple system atrophy [71]. Microglia and astrocytic disorders are reported in PD pathogenesis [72,73,74]. Microglia and astrocytes are classified as proinflammatory and neurotoxic M1 microglia and A1 astrocytes, and anti-inflammatory and neuroprotective M2 microglia and A2 astrocytes [75]. The persistence of MI-type microglia can contribute to the pathogenesis of neuroinflammatory disorders, including PD [75,76]. Microglial activation in the midbrain is seen in the early stages of PD pathogenesis (less than 2.5 years), whereas in later stages of the disease, increased microglial activation is seen in the SN, basal ganglia, cortex, and pons. These areas of increased microglial activation also correspond to the areas of PD pathology, as reported previously, indicating the role of microglial activation in PD [77,78]. In DLB, increased microglial activation is expressed in the putamen and cortical areas as well [71].

The present study demonstrates increased NFTs and APs in PDD brains. These APs and NFTs may be responsible for the increased activation of astrocytes and microglia in PDD brains, thereby increasing the expression of the GMF. Both local and systemic inflammatory factors play an important role in dementia [51,79]. High levels of proinflammatory mediators and the release of neurotoxic molecules in the brain directly accelerate the neurodegeneration process [17,18]. Previous studies have demonstrated that the presence of APs in the striatum is correlated with the severity of AD, and that the presence of these plaques may be associated with clinical dementia in patients with PD [16,80,81,82,83]. Studies have reported the implication of Aβ, tau, and NFTs in many neurological disorders, including PD [84,85]. The presence of APs has been shown to interrupt limbic connections, thereby disrupting striatal functions in patients with PD [83,86]. Our previous studies on AD demonstrated the increased expression of GMF immunoreactive glial cells in association with APs and NFTs in the entorhinal cortex in AD brains [87,88]. GMF-associated glial activation could contribute to PD pathogenesis. Future studies could include immunoblotting to confirm the marker for PDD, as well as single-cell and spatial transcriptomics to further confirm these findings.

## 4. Materials and Methods

### 4.1. PDD and Non-PDD Brains

Neuropathologically confirmed cases of PDD (*n* = 5) and age and gender-matched non-PDD (*n* = 6) controls were used in this study. Control brains had no neurological diseases or memory loss. These brains were collected at the University of Iowa (Iowa City, IA, USA) under the Deeded Body Program, as we have reported previously [42,89,90]. PDD patients have progressive motor disorder-associated symptoms and dementia in the later stage. The blocks of striatum and midbrain containing the SN were fixed with 4% paraformaldehyde solution and then maintained with 30% sucrose in phosphate-buffered saline (PBS) solution until the sections were ready to be cut (40 μm sections) on the sliding freezing microtome. The Institutional Review Board (IRB) of the University of Iowa and the University of Missouri (Columbia, MO, USA) Colleges of Medicine approved this study.

The workflow of the experiments includes histology–thioflavin-S fluorescence staining–immunohistochemistry–immunofluorescence staining–quantification.

### 4.2. Thioflavin-S (Thio-S) Fluorescence Staining

Midbrain containing SN and striatum sections of human PDD were stained with 1% thioflavin-S (Sigma, St. Louis, MO, USA) to identify APs in the brain, as we have reported previously [87]. Stained sections were examined under a Nikon fluorescence microscope to detect APs and NFTs in the SN region of the midbrain and striatum of human PDD and non-PDD brains.

### 4.3. Immunohistochemistry (IHC)

Immunohistochemistry was performed as previously described [91,92]. Briefly, antigen retrieval was performed in free-floating sections of the midbrain containing SN and striatum by boiling at 90 °C in citrate buffer solution (0.01 M, pH 6.0) for 10 min. Sections were treated with 0.3% hydrogen peroxide (H2O2) in PBS for 20 min to remove endogenous peroxidase activity. We employed the avidin–biotin–horseradish peroxidase complex (ABC, Vector Labs, Burlingame, CA, USA) standard staining technique for immunohistochemistry. Free-floating sections of PDD and non-PDD midbrain and striatum were used for IHC with tyrosine hydroxylase (TH) antibody (1:500 dilutions; Chemicon International, Temecula, CA, USA) to detect the loss of DA neurons and fibers. GMF antibody (1:200 dilutions, Proteintech, Rosemont, IL, USA) and polyclonal antibody to the glial fibrillary acidic protein (GFAP; 1:100 dilutions; Abcam, Cambridge, MA, USA) was used to identify the reactive astrocytes, and a polyclonal antibody to ionized calcium-binding adaptor molecule 1 (Iba1, 1:500 dilutions Wako Chemicals, Richmond, VA, USA) was utilized for the identification of activated microglia, and anti-alpha-synuclein (rabbit polyclonal, 1:500 dilutions, Chemicon, Temecula, CA, USA) and anti-ubiquitin rabbit polyclonal (1:500 dilutions, Dako, Santa Clara, CA, USA) antibodies were used to detect the Lewy bodies. Beta-amyloid monoclonal antibody (6E10, 1:1000, Covance, Dedham, MA, USA) was for APs, and phosphorylated tau-AT8 antibody (1:1000, Thermoscientific; Rockford, IL, USA) was used for NFTs. Sections were kept in blocking solution (5% normal goat serum, 3% bovine serum albumin, and 0.3% Triton X100 in PBS) for 1 h. After blocking, these sections were incubated with primary antibodies for 48 h at 4 °C followed by incubation with the appropriate biotinylated secondary antibodies (Vector Labs, 1:200) for 2 h at room temperature. These sections were then rinsed three times with PBS for 10 min each and incubated with the ABC staining kit (Vector Labs) solution diluted at 1:2000 in PBS for 1 h at room temperature. Peroxidase labeling was developed by incubating the sections with Vector SG peroxidase (HRP) substrate solution (Vector Labs) to detect the bound antibody–antigen enzyme complex, which results in a blue–grey color. For the negative staining control, the sections were processed without incubating with the primary antibodies. The sections were mounted on slides and dried at room temperature overnight. Slides were then dehydrated with ethanol, cleared in xylene, and coverslipped with Permount mounting medium (Fisher Scientific, Pittsburgh, PA, USA).

### 4.4. Immunofluorescence Staining

Immunofluorescence labeling was performed in coronal sections of human PDD brain areas for APs with beta-amyloid monoclonal antibody (6E10, Covance, Dedham, MA) and AT8 antibody (Thermoscientific, Rockford, IL, USA) for NFTs. After antigen retrieval with citrate buffer solution (0.01 M, pH6.0) at 90 °C for 10 min for free-floating sections, the sections were treated with 0.3% H_2_O_2_ in PBS for 20 min to block endogenous peroxidase reaction. Sections were incubated with blocking solution of 5% normal goat serum, 3% bovine serum albumin, and 0.3% Triton X 100 in PBS for 1 h. After blocking, sections were incubated with the primary antibodies of beta-amyloid monoclonal antibody (6E10, 1:1000 Covance, Dedham, MA, USA) and phosphorylated tau-AT8 antibody (1:1000 dilution, Thermoscientific, Rockford, IL, USA) for NFTs. After washing with PBS, the sections were incubated with secondary antibody goat anti-mouse Alexa Fluor 568 for 1 h at room temperature. The sections were washed with PBS and mounted on the slides. For control staining, the primary antibodies were omitted. The sections were analyzed under a Nikon standard fluorescence microscope.

### 4.5. Statistical Analysis

All the results are presented as mean ± SEM. Statistical analysis of the data was performed using Student’s independent *t*-test to determine statistically significant differences between the groups by using GraphPad InStat version 3 software (GraphPad Software, Boston, MA, USA). A *p*-value of less than 0.05 was considered statistically significant.

## 5. Conclusions

In conclusion, our present results demonstrate that the increased expression of α-synuclein, ubiquitin, NFTs, and Aps, along with the GMF and a reduced number of TH^+^ neurons, was found in the SN of PDD brains. An increased number of activated astrocytes and microglia colocalized with the GMF were seen in PDD brains. In addition, we also found that Tau proteins and NFTs with 6E10/APs-stained neuronal cells were increased in the SN, striatum, entorhinal cortex, hippocampus, and temporal cortex. The increased expression of the GMF could play an important role in the pathogenesis of PD and PDD. We conclude that the enhanced expression of the GMF correlates with reactive astrocytosis, along with the increased activation of microglial cells in the presence of APs and NFTs in PDD brains secondary to the occurrence of dementia-related pathology.

## Figures and Tables

**Figure 1 ijms-25-01182-f001:**
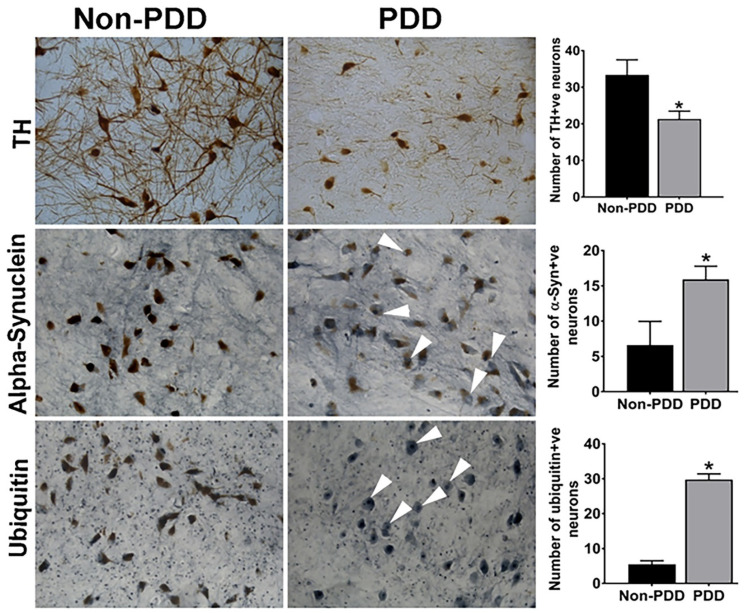
Expression of TH, α-synuclein (α-Syn), and ubiquitin in tissue sections of SN from PDD and non-PDD brains. Representative images show the immunoreactivity for TH-positive neurons, immunoreactive Lewy bodies labeled with α-synuclein (arrowheads), and ubiquitin (arrowheads) in the SN of PDD brains. Bar graphs show that the number of neuromelanin-pigmented TH^+^ neurons was significantly reduced, while α-Synuclein and ubiquitin^+^ neurons were significantly increased in PDD brains compared with non-PDD brains. More Lewy bodies were observed in the dopaminergic neurons in the SN of PDD brains. The results are presented as mean ± SEM. Statistical analysis of the data was performed using Student’s independent *t*-test with GraphPad InStat version 3 software. * *p* < 0.05 was compared with non-PDD and considered statistically significant. Magnification = 200×.

**Figure 2 ijms-25-01182-f002:**
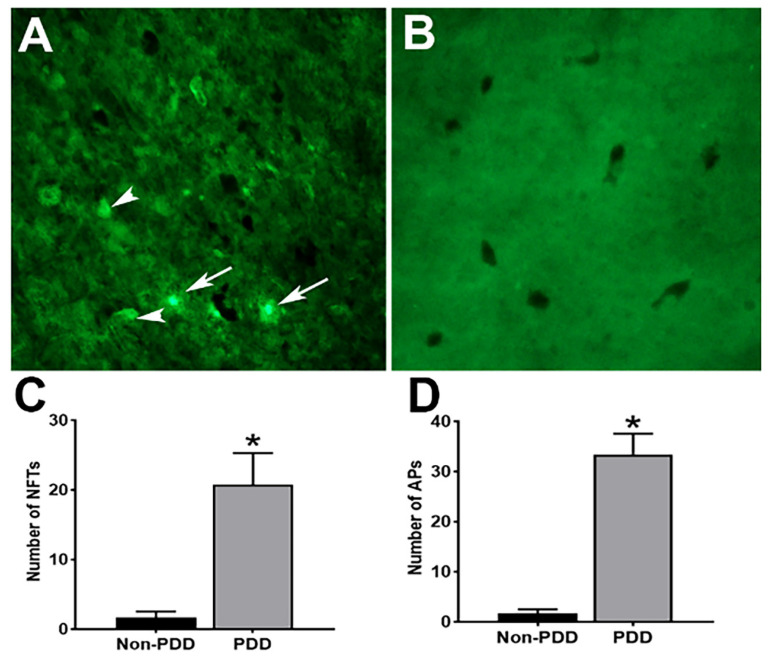
Expression and detection of APs and NFTs in the SN of PDD brains. Representative images of thioflavin-S (green fluorescence) stained with APs (white arrows) and NFTs (white arrowheads) are shown in the SN region of the PDD brain (**A**). Non-PDD brains did not show either NFTs or APs (**B**). Bar graphs show the thioflavin-S-stained number of NFTs (**C**), and the number of APs (**D**) was significantly increased in the PDD brain compared with the non-PDD brain. The results are presented as mean ± SEM. Statistical analysis of the data was performed using Student’s *t*-test by using GraphPad InStat version 3 software. * *p* < 0.05 was compared with non-PDD and considered statistically significant. Magnification = 200×.

**Figure 3 ijms-25-01182-f003:**
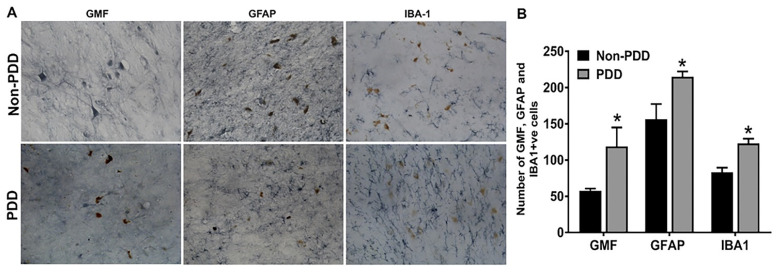
Expression of GMF-associated activated astrocytes and microglia in the SN of PDD and non-PDD brains. Representative images display immunoreactivity for GMF, GFAP, and Iba-1 in the SN of PDD brains (**A**). GFAP immunoreactive astrocytes are blue–gray, and Iba-1 positive activated microglia (blue–gray color) were observed near brown neuromelanin-containing dopaminergic neurons in the SN. Bar graphs show that the number of GFAP, IBA1, and GMF co-expressed cells was significantly increased in PDD brains compared with non-PDD brains (**B**). Results are presented as mean ± SEM. Statistical analysis of the data was performed using Student’s *t*-test by using GraphPad InStat 3. * *p* < 0.05 was compared with non-PDD and considered statistically significant. Magnification = 200×.

**Figure 4 ijms-25-01182-f004:**
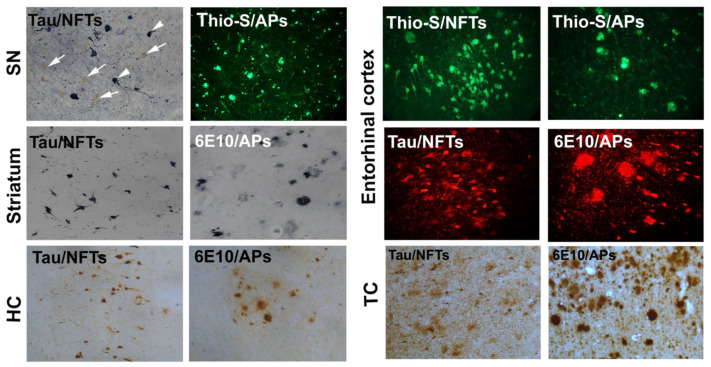
NFTs and APs pathology in the SN, striatum, entorhinal cortex, hippocampus (HC), and temporal cortex (TC) of PDD brains. Representative images show tau-stained NFTs (blue–gray color, arrowheads) and melanin-containing neurons (brown color, arrows) in the SN of PDD brains. Thioflavin-S (Thio-S) histochemical staining displays APs (green color, arrowheads) in the SN. We also show Tau immunostained NFTs in the striatum (blue–gray color, arrowhead) and 6E10 antibody immunolabeled APs in the striatum (blue–gray color, arrowheads) of PDD brains. Thioflavin-S-stained NFTs and APs and Tau (AT8) immunofluorescence labeling of NFTs (red) and 6E10 antibody-labeled APs (red) were observed in the entorhinal cortex. Tau antibody immunostained NFTs (brown color) and 6E10 antibody immunostained APs (brown color) in the hippocampus and temporal cortex of PDD brains. Magnification = 200×.

## Data Availability

Data will be provided upon reasonable request.

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
