# Peer review of "Parkinson’s Disease Dementia Patients: Expression of Glia Maturation Factor in the Brain"

_ijms, 2024, doi:10.3390/ijms25021182_

Round 1

Reviewer 1 Report

Comments and Suggestions for Authors

My suggestions

1. I would explain the PDD brains a little bit more in detail. What kind of symptoms did these patients have? What was their age of onset and age of death? Were the non-PDD control brains from healthy controls or did these individuals have some other kind of disease?

2. I recommend adding a workflow figure of experiments in the Methods section.

3. In the introduction, the authors may explain some genetic factors that may be associated with both Alzheimer's disease and Parkinson's disease.

4. Are there any reports available on the GMF impact of any other neurodegenerative diseases other than PD?Was  GMF examined in AD patients?

5. Was there any association between GMF localization with Tau aggregates and Lewy bodies?

6. Figure 4 may be uploaded in a bigger size. 

Author Response

point-by-point response uploaded

Reviewer 2 Report

Comments and Suggestions for Authors

These workers have previously detected expression of glia maturation factor (GMF) by glia and neurones in Alzheimer’s (AD) and Parkinson’s (PD) disease brains, after traumatic brain injury (TBI), and demonstrated its pro-inflammatory role in experimental autoimmune encephalomyelitis (EAE). Here, they have investigated GMF expression in five human PDD compared with six age-matched non-PDD brains. Using immunohistochemistry and thioflavin S, the expression pattern of GMF protein was compared with patterns of glial fibrillary acidic protein (GFAP) reactive astrocytes, loss of tyrosine hydroxylase (TH), alpha-synuclein, and ubiquitin by degenerating dopamine neurones, amyloid plaques (AP) and neurofibrillary tangles (NFT) load in the substantia nigra (SN) and striatum of PDD and non-PDD brains. The PDD cases showed a large number of GMF-positive, GFAP-reactive astrocytes within the SN and striatum. Raised levels of GMF were seen in glial cells surrounding APs, and NFTs in the SN and striatum of both PDD and non-PDD patients. It is concluded that the majority of GMF-positive immunoreactive glial cells were GFAP-reactive astrocytes and that GMF may be involved in the pathogenesis of PDD.

This study extends the previous studies from this group on the cellular expression of GMF in neurodegenerations and neuroinflammation. I have some comments:

First, it is not clear what diagnoses the non-PDD controls had – were they PD without dementia  or did they die from unrelated causes?

Second, did the GMF antibody detect both forms of this protein – both beta and gamma?

Third, as GMF expressing glia and raised GFAP are found in both AD and PD,  are the increased levels seen in PDD simply a reflection of the dual pathology present?

Fourth. although increased GMF expression is seen in PDD compared to non-PDD cases, no evidence is presented here that GMF plays a specific role in the pathogenesis of PDD.

In summary, the report is of interest but some clarification is needed as it stands.

Author Response

point-by-point response uploaded

Reviewer 3 Report

Comments and Suggestions for Authors

The authors analyzed 5 PDD and 6 non-PDD and found a majority of GMF-positive immunoreactive glial cells were co-localized with GFAP-reactive astrocytes. They concluded that GMF may be strongly involved in the pathogenesis of PDD patients. However, since only a few samples were done, more evidence should be provided with additional experiments to verify their finding.

1. Immunoblotting needs to be further performed and confirm the marker from PDD cases

2. The single-cell and spatial transcriptomics analysis that would help for their further confirmation (e.g: MERSCOPE)

Author Response

point-by-point response uploaded

Round 2

Reviewer 1 Report

Comments and Suggestions for Authors

Authors fulfilled my suggestions.

Author Response

response uploaded

Reviewer 3 Report

Comments and Suggestions for Authors

Thanks for addressing almost concerns. However, the Immunoblotting to verify the markers on the patient and control must be further performed to verify their finding since only a few samples were tested.  

Author Response

Response uploaded

Round 3

Reviewer 3 Report

Comments and Suggestions for Authors

Thanks for the revision again. Despite my concern is still not properly addressed due to no more samples, hope to see your upcoming results to verify the conclusions.